# Spectral Fingerprinting of Methane from Hyper-Spectral Sounder Measurements Using Machine Learning and Radiative Kernel-Based Inversion

Wan Wu [1],*, Xu Liu [1], Xiaozhen Xiong [1], Qiguang Yang [2], Lihang Zhou [3], Liqiao Lei [2], Daniel K. Zhou [1] and Allen M. Larar [1]

1 NASA Langley Research Center, Hampton, VA 23681, USA; xu.liu-1@nasa.gov (X.L.); xiaozhen.xiong@nasa.gov (X.X.); daniel.k.zhou@nasa.gov (D.K.Z.); allen.m.larar@nasa.gov (A.M.L.)
2 ADNET Systems Inc., Bethesda, MD 20817, USA; qiguang.yang@nasa.gov (Q.Y.); liqiao.lei@nasa.gov (L.L.)
3 NOAA JPSS Program Office, Silver Spring, MD 20706, USA; lihang.zhou@noaa.gov
* Correspondence: wan.wu@nasa.gov

**Abstract:** Satellite-based hyper-spectral infrared (IR) sensors such as the Atmospheric Infrared Sounder (AIRS), the Cross-track Infrared Sounder (CrIS), and the Infrared Atmospheric Sounding Interferometer (IASI) cover many methane ($CH_4$) spectral features, including the ν1 vibrational band near 1300 cm$^{-1}$ (7.7 μm); therefore, they can be used to monitor $CH_4$ concentrations in the atmosphere. However, retrieving $CH_4$ remains a challenge due to the limited spectral information provided by IR sounder measurements. The information required to resolve the weak absorption lines of $CH_4$ is often obscured by interferences from signals originating from other trace gases, clouds, and surface emissions within the overlapping spectral region. Consequently, currently available $CH_4$ data product derived from IR sounder measurements still have large errors and uncertainties that limit their application scope for high-accuracy climate and environment monitoring applications. In this paper, we describe the retrieval of atmospheric $CH_4$ profiles using a novel spectral fingerprinting methodology and our evaluation of performance using measurements from the CrIS sensor aboard the Suomi National Polar-orbiting Partnership (SNPP) satellite. The spectral fingerprinting methodology uses optimized CrIS radiances to enhance the $CH_4$ signal and a machine learning classifier to constrain the physical inversion scheme. We validated our results using the atmospheric composition reanalysis results and data from airborne in situ measurements. An inter-comparison study revealed that the spectral fingerprinting results can capture the vertical variation characteristics of $CH_4$ profiles that operational sounder products may not provide. The latitudinal variations in $CH_4$ concentration in these results appear more realistic than those shown in existing sounder products. The methodology presented herein could enhance the utilization of satellite data to comprehend methane's role as a greenhouse gas and facilitate the tracking of methane sources and sinks with increased reliability.

**Keywords:** methane; hyper-spectral infrared sounder measurements; CrIS; retrieval; spectral fingerprinting





## 1. Introduction

As a significant greenhouse gas with 28 times greater global warming potential than carbon dioxide [1], the recognition of the importance of monitoring methane ($CH_4$) in the atmosphere on a global scale has increased. Major absorption bands of $CH_4$ are located in the short-wave IR (centered at ~1.65 μm and ~3.3 μm) and mid-IR (centered ~7.7 μm) spectral regions, respectively. Spaceborne instruments like the Scanning Imaging Absorption Spectrometer for Atmospheric Chartography (SCIAMACHY) aboard the European Space Agency (ESA)'s Environmental Satellite (ENVISAT), the Thermal and Near-infrared Sensor for Carbon Observation–Fourier Transform Spectrometer (TANSO-FTS) aboard the Greenhouse Gases Observing Satellite (GOSAT), and the Tropospheric Monitoring Instrument (TROPOMI) aboard the Sentinel-5 Precursor satellite (Sentinel-5P) measure the

absorption of solar radiation by $CH_4$ in the short-wave IR band and provide near-surface sensitivity [2–4]. Both the SCIAMACHY and GOSAT measure strong $CH_4$ absorption signals around 1.65 μm. The TROPOMI measures $CH_4$ mixing ratios using the absorption information from the Oxygen-A Band (760 nm) and the short-wave IR band (2.3–2.4 μm). GOSAT also provides measurements in the thermal infrared band between 5.5 and 14.3 μm, which is used to derive thermal IR $CH_4$ data [5,6]. Compared with short-wave IR observations, mid-wave IR spectral measurements in the 7.7 μm mid-IR band are more sensitive to mid to upper tropospheric $CH_4$. IR sounders like the AIRS, IASI, and CrIS all measure the absorption of thermal IR radiation in the 7.7 μm band and therefore can provide important complementary information needed for global $CH_4$ monitoring.

Existing IR sounders have limited measurement sensitivity and instrument spectral resolution. Consequently, fully resolving weak $CH_4$ signals becomes difficult, impacting the accuracy of both total concentration and vertical profile distribution of the retrieved atmospheric $CH_4$. $CH_4$ absorption lines in the mid-IR band overlap with those of other gases, such as water vapor and nitrous oxide ($N_2O$), further introducing difficulties in isolating and accurately quantifying $CH_4$ concentrations. Retrieval studies using the AIRS and IASI have shown that the retrieved upper tropospheric $CH_4$ can easily have a bias error ranging from 1 to 4% [7,8]. Studies have shown inconsistencies between the $CH_4$ profiles retrieved from AIRS and GOSAT TIR measurements, as well as inconsistencies between the $CH_4$ profiles retrieved from AIRS and IASI measurements [9,10]. And the global spatial distribution of these $CH_4$ retrieval products remains to be validated.

$CH_4$ retrieval studies using hyper-spectral IR sounders measurements have been mostly based on the optimal estimation methodology (OEM) following Rodger's formulism [11]. A priori knowledge of $CH_4$ is critically needed to complement information content from the IR sounder measurements. Xiong et al., used latitudinal-dependent $CH_4$ first-guess profiles derived from the monthly averaged results of in situ aircraft observations, ground-based flask network measurements, satellite observations, and the atmospheric transport model TM3 [7]. García et al., retrieved $CH_4$ from IASI measurements using mean profiles from WACCM (Whole 30 Atmosphere Community Climate Model-version 5, https://www2.acom.ucar.edu/gcm/waccm), averaged on a 1.9° × 2.5° grid for the 2004–2006 period, as the first-guess profiles [12]. An ad hoc Tikhonov–Philips slope constraint is used to maintain the vertical shape of the $CH_4$ profiles during the retrieval process. De Wachter et al. also used the a priori profile and the covariance matrix constraint derived from WACCM, but a single global climatological a priori was used in their study [8]. Siddans et al. used a fixed value of 1.75 ppmv in the troposphere as the a priori state and two years of zonal mean values from the TOMCAT chemical transport model to construct a priori error statistics for IASI $CH_4$ retrievals [13]. Razavi et al., used the a priori derived from the "Laboratoire de Météorologie Dynamique" global climate model that constraints north–south latitudinal gradients of $CH_4$ distribution [14].

OEM usually assumes a Gaussian distribution of the possible solution around its a priori state, as well as a liner relationship between the change in $CH_4$ concentration from the a priori state and the associated change in observed radiance. Considering the complexity of the inverse relationship to be established for $CH_4$ retrieval, Crevoisier et al. used a neural network-based non-linear inference method to retrieve $CH_4$ from IASI observations [15]. The neural network-based approach, in theory, can be used to represent a non-linear training-prediction relationship, but the prediction accuracy largely depends on the representativeness of the training samples. The neural network-based approach also lacks the error estimation that is provided by the OEM-based scheme. The approach was trained using simulated data. The difference between simulated radiances and the observations was simply addressed via a bias correction based on one year of data over the tropical region. Considering the scene-dependent nature of the simulation error, large uncertainty between the simulation and the observation likely remains and inevitably contributes to the retrieval error.

In this paper, we present a novel $CH_4$ retrieval methodology based on the spectral fingerprinting approach. Such an approach utilizes spectral information from IR sounder measurements to derive a scene-dependent a priori state and corresponding constraint for each individual $CH_4$ retrieval. Well-defined a priori information will greatly improve the linearity of the inverse relationship between the atmospheric $CH_4$ and the spectral radiances observed, as well as the Gaussian distribution characteristics of the solution around the a priori state. The spectral fingerprinting scheme is based on a pre-constructed database that comprises an ample set of representative reference states, along with their corresponding radiative kernels. In this approach, a clustering method based on machine learning is employed to stratify and identify the scene-dependent a priori state within the pre-constructed database. Spectral radiances serve as the predictors. The a priori state is identified via radiance spectral matching, and the corresponding radiative kernel is then used to find the fingerprinting solution via an OEM-based linear inversion scheme. Details about the spectral fingerprinting scheme will be introduced, with in-depth technical insights into the construction of the fingerprinting database also being provided. The application of the fingerprinting methodology on SNPP CrIS observations will be demonstrated with the derived $CH_4$ profiles validated using both $CH_4$ reanalysis data from the Copernicus Atmosphere Monitoring Service (CAMS) and in situ measurement data from the Atmospheric Tomography Mission (ATom) [16].

The $CH_4$ fingerprinting algorithm study presented herein contributes to the efforts to improve the single field-of-view sounder atmospheric product (SIFSAP) [17]. As a novel algorithm developed to complement other sounder products, such as CLIMCAPS and AIRS version 7 [18,19], the SiFSAP system produces hyper-spectral IR sounder Level 2 data at an instrument-native spatial resolution and provides high-accuracy spectral fitting to the top of the atmosphere (TOA) sounder observations by directly simulating cloud scattering. In order to maximize the information content from the measurements (i.e., minimized use of a priori information), the global climatology-based a prior constraint is used in the SiFSAP retrieval algorithm. We purposely designed the SiFSAP algorithm using relaxed a priori global climatology so that the retrieval results are more sensitive to the small climate signals caught by the TOA spectral radiance observations. High-quality SiFSAP Level 2 products of atmospheric temperature, water vapor, and other trace gases such as ozone ($O_3$) have been used for various dynamics and climate studies [20]. However, the uncertainty in $CH_4$ data at a localized, instantaneous scale can be potentially large because the $CH_4$ information provided by the measurements can be very limited for a significant percentage of cases. The fingerprinting algorithm can be used to generate a high-quality first-guess and scene-dependent a priori covariance constraint to improve the $CH_4$ retrieval in the SiFSAP system for near-real-time applications. It will be implemented to produce the next version of SiFSAP.

## 2. The Spectral Fingerprinting Methodology

The spectral fingerprinting methodology has been widely used for the characterization and quantification of biological materials, chemical components, mineral analysis, and remote sensing [21–24]. The concept is based on the fact that the spectral feature of a measured signal can be used for the constitutive component analysis of target samples. The analysis usually involves the classification of spectral features associated with known constituents so that the constitutive component can be identified by characterizing the similarity between the measured signal and the spectral features of a prescribed constituent. When the measured spectral signal *r* of a target sample matches a known reference spectrum, the constituents of the target sample *x* are instantly identified using the reference database. In broad terms, spectral fingerprinting integrates both spectral classification and spectral matching procedures. Recent corresponding research in the field of remote sensing has predominantly centered on the application of spectral fingerprinting in the analysis of hyperspectral images [25–27].

If the target signal is composed of multiple constituents, the spectral fingerprinting also involves the decomposition of the total measured signal into its different spectral components of individual constituents. The technique has been used for the optimal detection and attribution of climate change signals in the outgoing spectra of TOA radiances [28–32]. In those studies, the spectral fingerprints are the anomalies of outgoing spectral radiances, and the constituents are the changes in the TOA spectra that are associated with different feedbacks and forcings. The attribution of those spectral fingerprints to the change in essential climate variables involves the use of radiative kernels and a linear inversion scheme. The spectral fingerprinting scheme can be expressed as:

$$\Delta r = \mathbf{K}\Delta x + \varepsilon, \tag{1}$$

where $\Delta r$ is the spectral fingerprints, $\mathbf{K}$ is the radiative kernel, $\Delta x$ is the change in the geophysical variables, and $\varepsilon$ is the fingerprinting residual term. $\Delta x$ can be derived from the spectral fingerprints as follows:

$$\Delta x = \left(\mathbf{K}^T \mathbf{\Sigma}^{-1}\mathbf{K} + S_a^{-1}\right)^{-1}\mathbf{K}^T\mathbf{\Sigma}^{-1}\Delta r, \tag{2}$$

where $\mathbf{\Sigma}$ is the covariance of the residual term $\varepsilon$, and $S_a$ is the covariance constraint of $\Delta x$. In the satellite-based remote sensing of trace gases, spectral fingerprints are established as variations in the observed spectral radiances $r$ with respect to the reference spectrum $r_0$. These variations are exclusively attributed to the changes in the trace gas profile $x$ from the reference profile $x_0$. The radiative kernel $\mathbf{K}$ defines the linear relationship between $\Delta r$ and $\Delta x$.

The spectral fingerprinting of $CH_4$ using the IR sounder measurements includes both classification and one-step linear inversion procedures. The classification analysis is carried out to build a predictive model based on a reference database that includes the representative radiance spectra, the collocated $CH_4$ profile data, and corresponding radiative kernels. The classification of IR spectral measurements is based on their spectral characteristics associated with the $CH_4$ absorption. A given IR sounder observation can therefore be automatically assigned to one of the predefined classes. The $CH_4$ profile $x_0$, the corresponding spectrum $r_0$, the radiative kernel $\mathbf{K}$, and the fingerprinting residual covariance $\mathbf{\Sigma}$ of the assigned class are then used in the subsequent inversion procedure. The final solution $x$ is given by adjusting $x_0$ to account for the small spectral difference $\Delta r$ between the measurements $r$ and $r_0$:

$$x = x_0 + \left(\mathbf{K}^T\mathbf{\Sigma}^{-1}\mathbf{K} + S_a^{-1}\right)^{-1}\mathbf{K}^T\mathbf{\Sigma}^{-1}(r - r_0), \tag{3}$$

where the radiative kernel $\mathbf{K}$ is derived from the radiative transfer calculation as the Jacobian, i.e., the derivative of the TOA radiance with respect to the changes in $x$, for the assigned class.

$$\mathbf{K} = {dr}/{dx}|_{x=x_0}. \tag{4}$$

The formulas shown in Equations (2) and (3) follow the standard OEM scheme. Further details regarding the establishment of $x_0$, $r_0$, and $\mathbf{K}$, along with the construction of covariance matrices $\mathbf{\Sigma}$ and $S_a$, will be elaborated upon in Section 3.2.2.

## 3. Implementation of the Spectral Fingerprinting Scheme

### 3.1. Optimization of Spectral Information

Strong absorption lines of $H_2O$ and $N_2O$ exist in the spectral region where $CH_4$ absorption lines are located. In order to reduce the interference from those two species to the $CH_4$ retrieval, we select IR sounder spectral channels that are relatively less sensitive to $H_2O$ and $N_2O$ and more sensitive to $CH_4$. Figure 1 shows sample Jacobians of $N_2O$, $H_2O$, and $CH_4$ in the 1210–1390 cm$^{-1}$ spectral region which illustrate the brightness temperature (BT) responses of unapodized CrIS spectra introduced by the changes in three trace gas species at different pressure levels, respectively. Nine CrIS channels (marked by blue dots

in Figure 1) that are sensitive to $CH_4$ but not to $N_2O$ are chosen. Those channels are selected to avoid the interference in the 1240–1320 cm$^{-1}$ spectral region, where $N_2O$ interference is strong. In order to mitigate the impact of broad spectral features such as the water continuum absorption, cloud emission and scattering, and surface emissivity variation in the selected channels, we construct differential spectral signals by pairing those nine 'valley' channels that show strong responses to $CH_4$ absorption with adjacent 'shoulder' channels (marked by '□' in Figure 1). Table 1 lists the 'valley' and the 'shoulder' channels selected for $CH_4$ retrieval. Using the difference between the two sets of channels enhances the spectral signal contrast between $CH_4$ and $H_2O$ and therefore facilitates the identification of $CH_4$ fingerprints. To further enhance the information due to trace gas absorption, we normalize the $CH_4$ channel radiances by using the measurement from the atmospheric 'window' region. Using a radiance ratio from between the $CH_4$ absorption channels and a 'window' channel (e.g., the CrIS channel at 900.625 cm$^{-1}$) can reduce the contributions from surface properties (emissivity and skin temperature) and clouds, which helps to better catch the spatial–temporal difference among sounder measurements mainly due to the change in $CH_4$ concentration. The spectral radiance $r$ (in Equation (3)) to be used for the fingerprinting can be formulated as

$$r = \frac{r_{valley} - r_{shoulder}}{r_{window}} \tag{5}$$

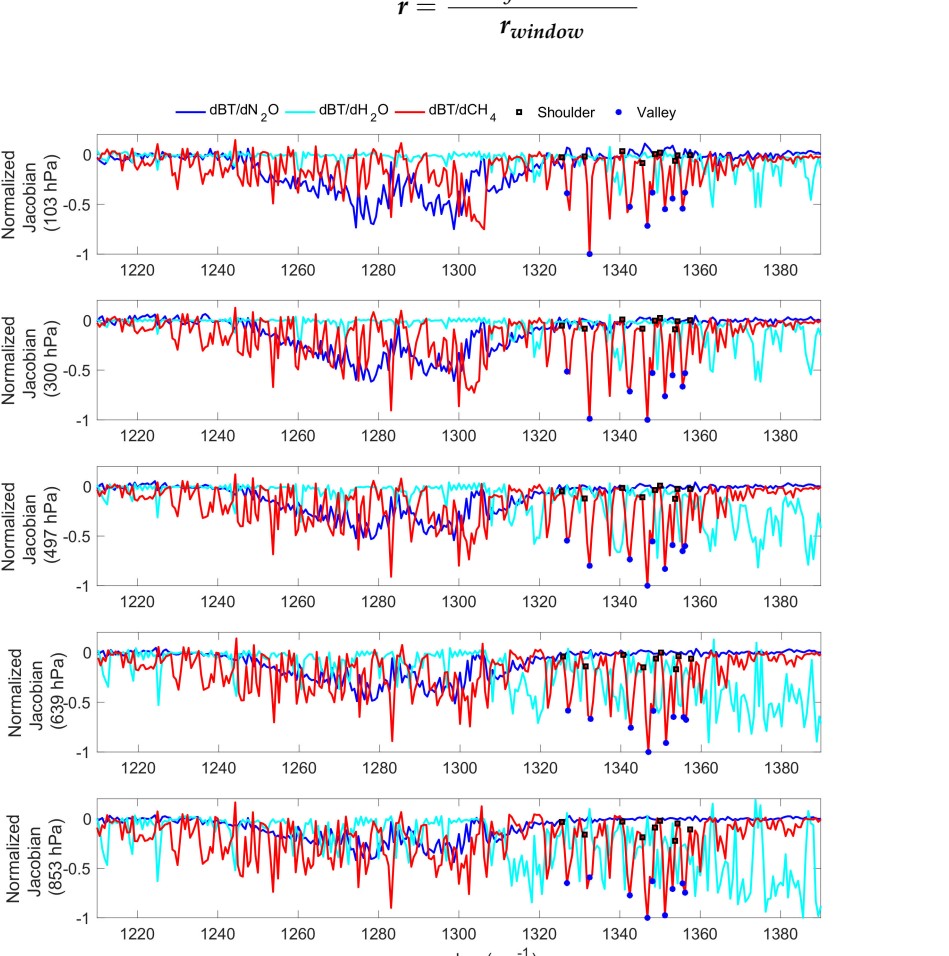

**Figure 1.** CrIS (unapodized) Jacobians of $CH_4$, $N_2O$, and $H_2O$ at different pressure levels simulated by PCRTM. The demonstrated Jacobians are normalized by dividing the values by the maximum value for each dataset to better highlight the spectral response difference. The red squares and blue dots mark the 'shoulder' and 'valley' channels selected for the spectral fingerprinting of $CH_4$, respectively.

**Table 1.** List of 'valley' and 'shoulder' channels selected for the spectral fingerprinting of $CH_4$ (highlighted in Figure 1).

|  | 'Valley' Channel $\mathcal{V}$ (cm$^{-1}$) | 'Shoulder' Channel $\mathcal{V}$ (cm$^{-1}$) |
|---|---|---|
| 1 | 1326.875 | 1325.625 |
| 2 | 1332.500 | 1331.250 |
| 3 | 1342.500 | 1340.625 |
| 4 | 1346.875 | 1345.625 |
| 5 | 1348.125 | 1348.750 |
| 6 | 1351.250 | 1350.000 |
| 7 | 1353.125 | 1353.750 |
| 8 | 1355.625 | 1354.375 |
| 9 | 1356.250 | 1357.500 |

*3.2. Classification Using the Reference Database*

3.2.1. Data Sources of the Reference Database

Spectral data are obtained from the CrIS Level 1B Full Spectral Resolution radiance data product [33]. CrIS is a Fourier transform infrared spectrometer with three measurement bands that cover long-wave IR (650–1095 cm$^{-1}$), mid-wave IR (1210–1750 cm$^{-1}$), and short-wave IR (2155–2550 cm$^{-1}$) spectral regions, respectively. Its spectral measurements from 1210 to 1400 cm$^{-1}$ provide the information that can be used for $CH_4$ retrieval. We used the results from the Carbon Tracker-$CH_4$ (CT-$CH_4$) system as the $CH_4$ reference profiles [34]. CT-$CH_4$ provides global-scale 3 h interval $CH_4$ mixing ratio profile data for each $3.0° \times 2.0°$ (longitude × latitude) grid cell. CT-$CH_4$ data are spatio-temporally interpolated to match the selected CrIS observations and saved in the reference database. The radiative kernels (**K** in Equation (4)) are simulated using the Principal Component-based Radiative Transfer Model (PCRTM) [35]. Except for $CH_4$, the other input parameters (e.g., temperature, water vapor, cloud, and surface properties) needed for the radiative kernel simulation are extracted from the SNPP CrIS SiFSAP [18].

The data In the reference database are matched to CrIS observations spanning six years (2016–2021). The objective is to establish a database with low redundancy that effectively captures the vertical and geographic distribution of $CH_4$ profiles from the original CT-$CH_4$ dataset, along with the corresponding spectral information from CrIS measurements. A stratified random sampling strategy is implemented for data selection. The CrIS observations are initially divided into 15 groups based on their relative across-track scan positions from nadir. Within each group, observations with a defined scan angle range are further organized by five-degree latitude zones. The data within each latitude zone are stratified based on surface pressure and $CH_4$ surface concentration values. Observations are then selected via random sampling, adhering to the joint probability distribution of the surface pressure and $CH_4$ surface concentration values for each latitude zone. To effectively capture the seasonal variations in $CH_4$ profiles, the selected samples are evenly distributed across each month. For each scan angle group, more than 400,000 observations are chosen and utilized to construct distinct reference databases. Each reference database includes spectral radiances (as defined in Equation (5)), CT-$CH_4$ profiles matched to the corresponding observations, radiative kernels, and associated surface pressure and latitude information.

3.2.2. Classification Based on Optimized Spectral Radiance Information

Building the inversion scheme defined by Equation (3) requires the use of a classification procedure to find the reference state and therefore its corresponding components $r_0$, $x_0$, and **K** for a given observation with a normalized differential spectral radiance vector $r$ (defined in Equation (5)). The procedure involves selecting a group of samples from the reference databases using nearest neighbor search. The Euclidean distance between $r$ and the sample values from the database is used as the distance metric. The scope of the search is limited to reference samples within a 20-degree latitude zone of the observation and with close surface pressure values (within $\pm 100$ hPa). Considering the search is carried out in a low dimension (nine channel pairs), we employ a K Dimensional-tree (KD-tree)

structure to partition the spectral radiance dataset, enabling highly efficient searches. The nearest neighbor search selects a dataset of $N$ ($N < 30$) samples that includes $CH_4$ profiles $\{x_1 \cdots x_N\}$, normalized differential spectra data $\{r_1 \cdots r_N\}$, and corresponding radiative kernels $\{\mathbf{K}^1 \cdots \mathbf{K}^N\}$. $\{x_1 \cdots x_N\}$ are expected to be distributed around the true $CH_4$ profile of the given observation. With $r_0$, $x_0$, and $\mathbf{K}$ being established using the mean values of the dataset, the residual dataset $\{\varepsilon_1 \cdots \varepsilon_N\}$ can be calculated as follows:

$$\varepsilon_i = r_i - \mathbf{K}(x_i - x_0). \tag{6}$$

The covariance terms ($\mathbf{\Sigma}$ and $S_a$ in Equation (3)) can therefore be derived as the covariance matrix of $\varepsilon$ and $x - x_0$, respectively.

### 3.3. Constraints on CH$_4$ Profiles via Parameterization

Similar to other retrieval algorithms based on the OEM scheme, as depicted in Equations (2) and (3), the $CH_4$ retrieval sensitivity can be quantified by the averaging kernel matrix, which is formularized as

$$\mathbf{A} = \left(\mathbf{K}^T \mathbf{\Sigma}^{-1} \mathbf{K} + S_a^{-1}\right)^{-1} \mathbf{K}^T \mathbf{\Sigma}^{-1} \mathbf{K}. \tag{7}$$

The dimension of $\mathbf{A}$ is determined by the number of variables used to represent the $CH_4$ profile. The number of independent observations provided by the OEM scheme can be estimated by the degrees of freedom (DOF), which is quantified as the trace of $\mathbf{A}$. In an ideal retrieval system, vertical profiles of $CH_4$ can be perfectly represented as the combination of independent components targeted by corresponding observations. However, information content from CrIS, AIRS, and IASI measurements is limited. The DOF value for an IR sounder-based $CH_4$ retrieval system is typically around 1.0 or even smaller [7,14]. Therefore, the number of independent components that can be obtained from $CH_4$ retrieval is expected to be very low. A well-designed parameterization scheme should optimize the choice of parameters in order to better represent the vertical profiles using a minimal number of parameters.

We have selected three representative parameters to represent the $CH_4$ profiles, taking into account the fundamental physical factors that influence the vertical distribution characteristics of $CH_4$ in the atmosphere. $CH_4$ undergoes oxidation by the hydroxyl radical (OH) in the upper troposphere. Above the tropopause, its mixing ratios decrease significantly due to additional chemical reactions with atomic oxygen and chlorine. Photolysis also contributes to $CH_4$ decomposition in the higher-altitude stratosphere. The vertical profiles of $CH_4$, represented by symbol $f$, can be approximated as a sigmoid-like function:

$$f(h) = \frac{S}{1 + e^{-\left(\frac{h-P}{n}\right)}}, \tag{8}$$

where $S$ defines the near-surface mixing ratio, $P$ defines the pressure level of the turning point where the $CH_4$ concentration goes from an accelerating decline to a decelerating decline, and $n$ determines the rate of decrement in the stratosphere. The fingerprinting solution $x$ defined in Equation (2) can therefore be represented as the state vector $[S, P, n]$. In this way, the change in $CH_4$ profiles is constrained to a solution defined by three parameters:

$$\Delta f(h) = \frac{\partial f}{\partial S} \Delta S + \frac{\partial f}{\partial P} \Delta P + \frac{\partial f}{\partial n} \Delta n. \tag{9}$$

with

$$\frac{\partial f}{\partial S} = \frac{1}{1 + e^{-\left(\frac{h-P_0}{n_0}\right)}}, \tag{10}$$

$$\frac{\partial f}{\partial P} = \frac{S_0}{n_0} \frac{e^{-\left(\frac{h-P_0}{n_0}\right)}}{\left(1 + e^{-\left(\frac{h-P_0}{n_0}\right)}\right)^2}, \tag{11}$$

$$\frac{\partial f}{\partial n} = \frac{S_0 P_0}{n_0^2} \frac{e^{-(\frac{h-P_0}{n_0})}}{\left(1 + e^{-(\frac{h-P_0}{n_0})}\right)^2},$$ (12)

and $S_0$, $P_0$, and $n_0$ are the parameters associated with the reference state vector $x_0$. The Jacobian $\mathbf{K}$ in Equation (2) can therefore be transformed from the profile domain to the parameter domain as $\mathbf{K}(x) = \tilde{\mathbf{K}}(f)\mathbf{T}$. The dimension of the transformation matrix $\mathbf{T}$ is $m \times 3$, where $m$ denotes the number of level (layers) quantities used to represent a $CH_4$ profile. The components of the $i$th row of the $\mathbf{T}$ matrix are $[\frac{\partial f(h_i)}{\partial S}, \frac{\partial f(h_i)}{\partial P}, \frac{\partial f(h_i)}{\partial n}]$. The use of a sigmoid-like function (Equation (8)) to represent vertical profile serves as a smoothing constraint. Consequently, $\Delta f$ must follow the vertical distribution patterns defined by Equations (10)–(12). This effectively prevents $f$ from deviating from a sigmoid-like vertical profile to a solution with a drastically different and unrealistic vertical structure in the linear inversion step (Equation (2)).

## 4. Results and Validation

The left panels in Figures 2 and 3 show two days' worth of SNPP CrIS $CH_4$ fingerprinting results (only descending orbital results are presented for simplicity). The fingerprinting results are compared with the global reanalysis dataset of atmospheric composition produced by CAMS. CAMS reanalysis provides sub-daily data interpolated to a regular $0.75° \times 0.75°$ lat/lon grid. The CAMS reanalysis is produced using 4-DVar data assimilation in ECMWF's Integrated Forecasting System (IFS) using a comprehensive inventory, climatological, and chemical modeling dataset to initialize and constrain $CH_4$ emissions, natural sources/sinks, and chemical sinks [36]. The accuracy of the CAMS $CH_4$ results have been assessed in the CAMS greenhouse gas technical note [37]. When comparing the global mean values of CAMS $CH_4$ with observations, the bias in the difference is generally small, with an uncertainty value of 0.01 ppm. Specifically, both the bias and the standard deviation values of the difference between the $CH_4$ tropospheric profiles and the NOAA AirCore data are below 0.05 ppm. The largest difference in surface and tropospheric column $CH_4$ between the CAMS results and the Total Carbon Column Observing Network (TCCON) data, with an averaged magnitude of up to 2.5%, is observed at mid- and high-latitude TCCON sites. The difference between CAMS results and data from the Network for the Detection of Atmospheric Composition Change (NDACC) is about 0.4% across all NDACC sites. CAMS reanalysis data are not recommended for investigating a local emission change or quantifying the changes in the atmospheric $CH_4$ growth rate, but these data can be used to characterize synoptic spatial variability and the seasonal cycle of $CH_4$ [37].

We compared the global distribution of the upper to middle tropospheric $CH_4$ volume mixing ratio (VMR) characterized by the CrIS fingerprinting results with the corresponding daily mean values from the CAMS reanalysis. The side-by-side comparisons illustrated in Figures 2 and 3 show good correlations between two results concerning the latitudinal gradient and several large-scale thermodynamic characteristics. Satellite-based CrIS observations for $CH_4$ are inevitability affected by cloud contamination. As a result, certain areas with high concentrations of $CH_4$ in the tropical region, as indicated in the CAMS reanalysis, are not accurately reflected in the fingerprinting results. The systematic differences between the two sets of results, most notably in the southern polar region, fall within the range of 1–3%. Nevertheless, the changes in $CH_4$ concentration from winter to summer in 2017, depicted through the difference between Figures 2 and 3, are consistent in both datasets. Figure 4 demonstrates the latitudinal change in $CH_4$ concentration at different altitudes. In Figure 4, the latitudinal variations in $CH_4$ concentration at different altitudes are illustrated. Consistent latitudinal patterns spanning from the Southern Hemisphere to the Arctic region can be observed in the daily results of the CAM reanalysis and CrIS fingerprinting. Both results also capture similar seasonal changes in $CH_4$ profiles. Generally, there is a positive tropospheric $CH_4$ increment in the Southern Hemisphere, contrasting with a negative increment in the Northern Hemisphere. The contrast between the $CH_4$ profiles on January 11 and July 9 reveals a significant increase in $CH_4$ concentration within the 100–200 hPa vertical region over Antarctica (90–60°S) and the northern mid-latitude zone (30–60°N).

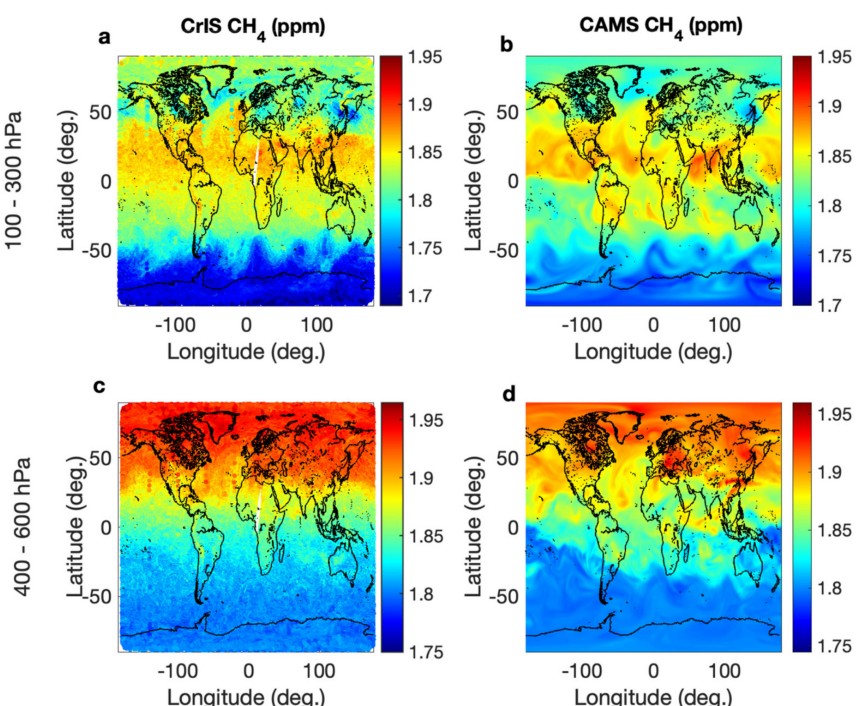

**Figure 2.** Comparison between daily CrIS CH$_4$ profile VMRs and those from CAMS reanalysis for 11 January 2017 in both the upper tropospheric region (100–300 hPa) and the middle tropospheric region (400–600 hPa). (**a**,**b**): averaged CH$_4$ volume mixing ratio in the 100—300 hPa latitudinal region derived from spectral fingerprinting results and from CAMS reanalysis). (**c**,**d**): averaged CH$_4$ volume mixing ratio in the 400—600 hPa latitudinal region derived from the spectral fingerprinting results and from the CAMS reanalysis.

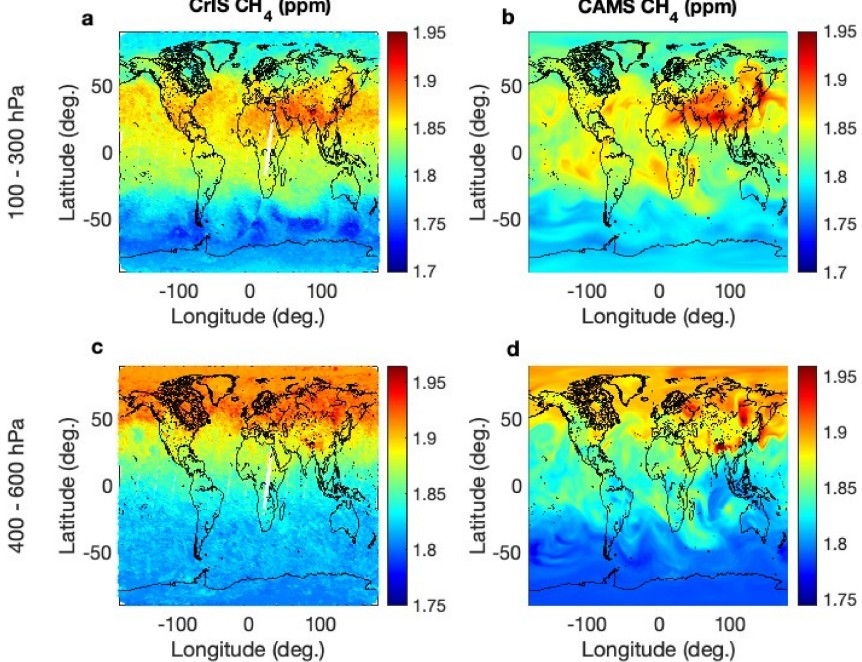

**Figure 3.** Comparison between daily CrIS CH$_4$ profile VMRs and those from CAMS reanalysis for 9 July 2017 in both the upper tropospheric region (100–300 hPa) and the middle tropospheric region (400–600 hPa). (**a**,**b**): averaged CH$_4$ volume mixing ratio in the 100—300 hPa latitudinal region derived from spectral fingerprinting results and from CAMS reanalysis). (**c**,**d**): averaged CH$_4$ volume mixing ratio in the 400—600 hPa latitudinal region derived from the spectral fingerprinting results and from the CAMS reanalysis.

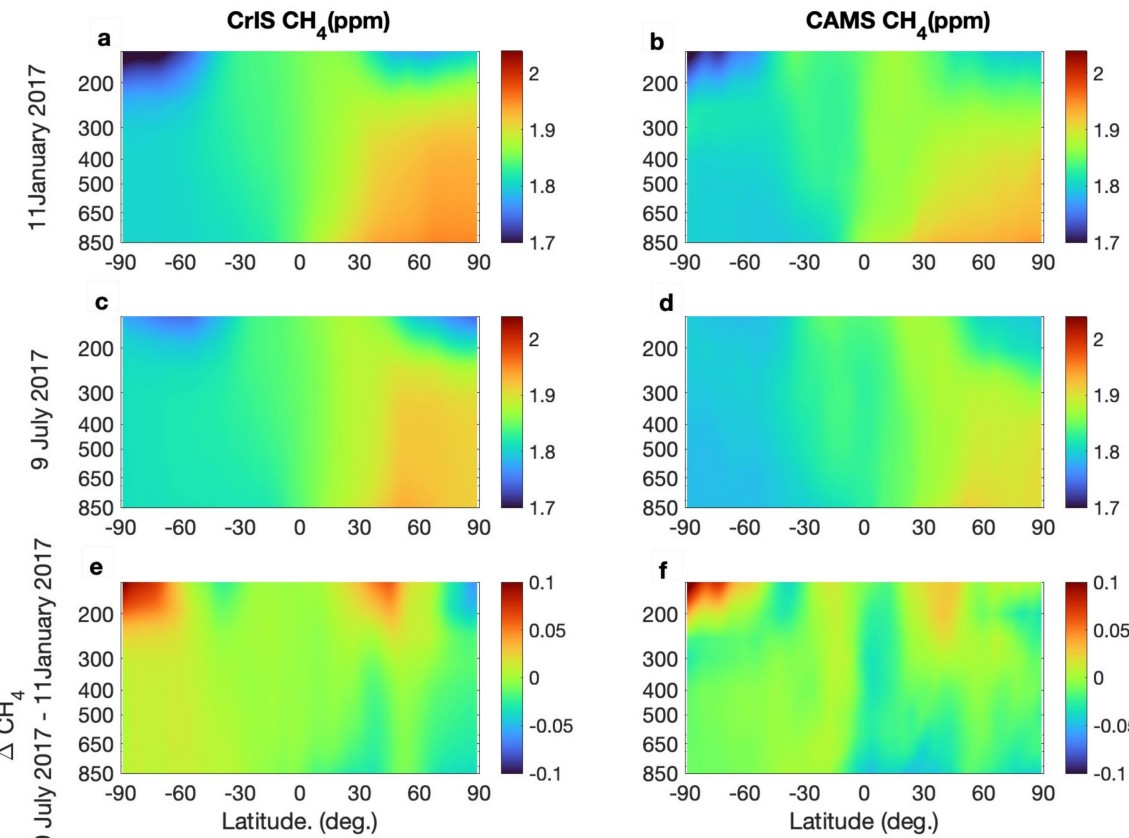

**Figure 4.** Latitudinal distribution of daily $CH_4$ profile (CAMS reanalysis versus CrIS fingerprinting). (**a**,**b**): $CH_4$ profiles of 11 January 2017 averaged to two-degree latitudinal zones; (**c**,**d**): $CH_4$ profiles of 9 July 2017 averaged to two-degree latitudinal zones. (**e**,**f**): Difference between $CH_4$ zonal mean profiles of 11 January 2017 and 9 July 2017. The daily zonal average has been computed for CrIS $CH_4$ data spanning all longitudes.

Hyperspectral sensors like the AIRS, IASI, and CrIS only have measurement sensitivity to $CH_4$ profiles within a limited vertical region. The AIRS is known to be sensitive to the upper troposphere (200–300 hPa) in the tropics and the middle troposphere (400–500 hPa) in the polar regions [7]. Similarly, studies on the IASI indicate its sensitivity to mid–upper tropospheric $CH_4$ [13–15]. CLIMCAPS exclusively provides $CH_4$ mass mixing ratios at 400 hPa, situated near the sensitivity peak defined by the algorithm and the spectral characteristics of the CrIS instrument. Despite the limited information from the measurements, the latitudinal distribution of the $CH_4$ profiles derived by fingerprinting SNPP CrIS measurements, as illustrated in Figures 2–4, generally agrees with the CAM reanalysis. This underscores the benefit of a scene-dependent a priori scheme that can be precisely constructed using the spectral fingerprinting methodology.

Under the fingerprinting scheme, the scene-dependent a priori information obtained through machine learning not only supplements the vertical information but also enhances the retrieval accuracy, particularly in areas where $CH_4$ signals are relatively weak. Physical retrieval algorithms like CLIMCAPS and AIRS version 7 are susceptible to errors in both the forward model and information from measurements. Addressing those errors becomes more challenging in regions covered by cloud and lacking thermal contrast. The fingerprinting methodology uses an a priori scheme obtained via machine learning to effectively constrain the impact of measurement errors in those regions. Figure 5 illustrates the latitudinal distribution of mid-tropospheric $CH_4$ (200–500 hPa) from the SNPP CrIS fingerprinting results of two days (11 January and 9 July 2017), along with the results from SNPP CLIMCAPS and AIRS version 7. Using CAM reanalysis and CT data as the reference, we can see that the daily latitudinal variations in $CH_4$ from AIRS version 7 and CLIMCAPS are

unrealistically large. The global-scale $CH_4$ concentration is underestimated in AIRS version 7. CLIMCAPS, on the other hand, overestimates $CH_4$ concentration over the subtropical region and underestimates it over the Antarctic region during the winter. In comparison with the operational sounder retrieval results, the fingerprinting results provide more reasonable estimates of both the magnitude of the $CH_4$ concentration and the latitudinal distribution on a global scale. These improvements in the accuracy of $CH_4$ data, which could significantly benefit studies that emphasize daily-scale geographical distributions.

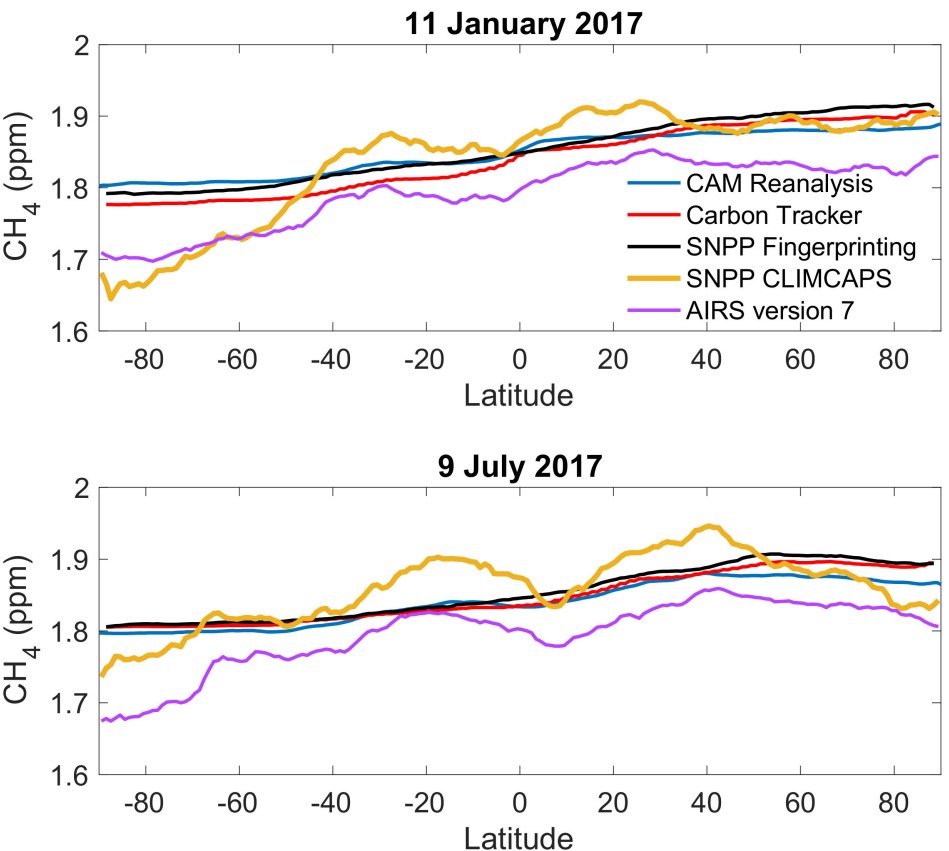

**Figure 5.** Inter-comparison between daily $CH_4$ concentration (mean values of 200–500 hPa) averaged over a one-degree latitudinal region obtained via different methods.

The SNPP CrIS $CH_4$ profiles were further validated using airborne measurement data. Three years' (2016–2018) worth of ATom data were used as the reference dataset. ATom observations provide continuous in situ measurements of $CH_4$ at various altitudes ranging from 1.2 km to 12 km. We began by selecting SNPP CrIS measurements that fall within a 12 h window of individual ATom observations. Subsequently, we generated collocated CrIS results through a two-dimensional space interpolation process. We excluded samples where the horizontal distance between an ATom observation and the nearest CrIS footprint exceeds 1 degree (~100 km). Each collocated sample's fingerprinting result, representing an individual $CH_4$ vertical profile, was then aligned with individual in situ results measured at different altitudes using vertical interpolation. ATom deployed flights over several months during a campaign year. The total number of collocated samples used for each year falls within the range of several hundred thousand. Figures 6–8 show the CrIS $CH_4$ results along with the collocated ATom in situ observations. It is evident that the fingerprinting results effectively capture the fluctuations in $CH_4$ concentration as the aircraft traversed different regions, collecting measurements at diverse altitudes.

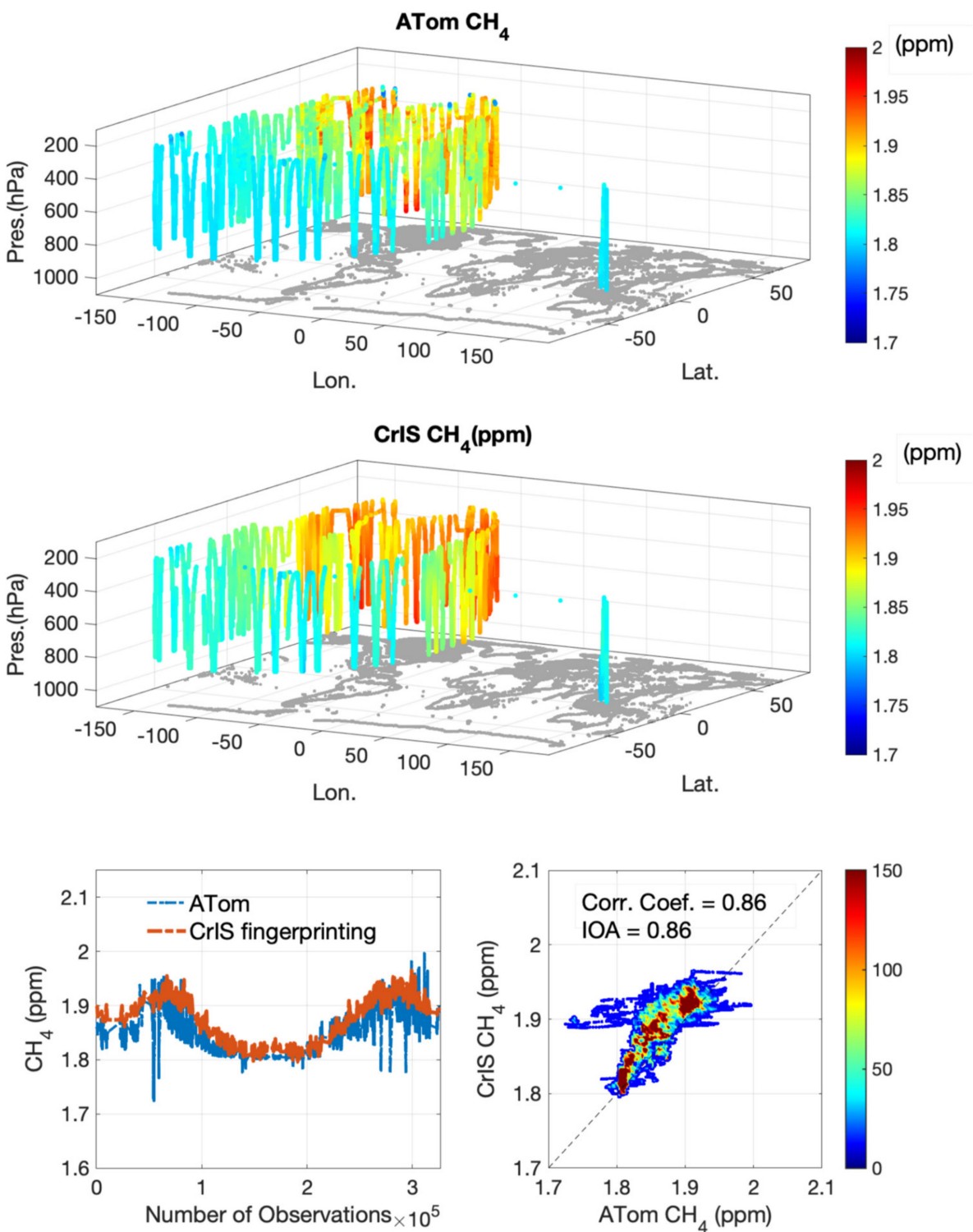

**Figure 6.** Upper panel: 3D illustration of CH$_4$ VMRs obtained via in situ ATom observations from 2016; middle panel: CH$_4$ VMRs retrieved from collocated SNPP CrIS observations; lower left panel: inter-comparison between two sets of results via an overplay plot; lower right panel: scatter plot of CrIS versus Atom CH$_4$ VMRs for all samples measured at a wide range of height and geolocations. The correlation coefficient between the fingerprinting results and Atom observations based on linear regression is displayed, along with the index of agreement (IOA) value. The color bar illustrates the number density of the collocated observations.

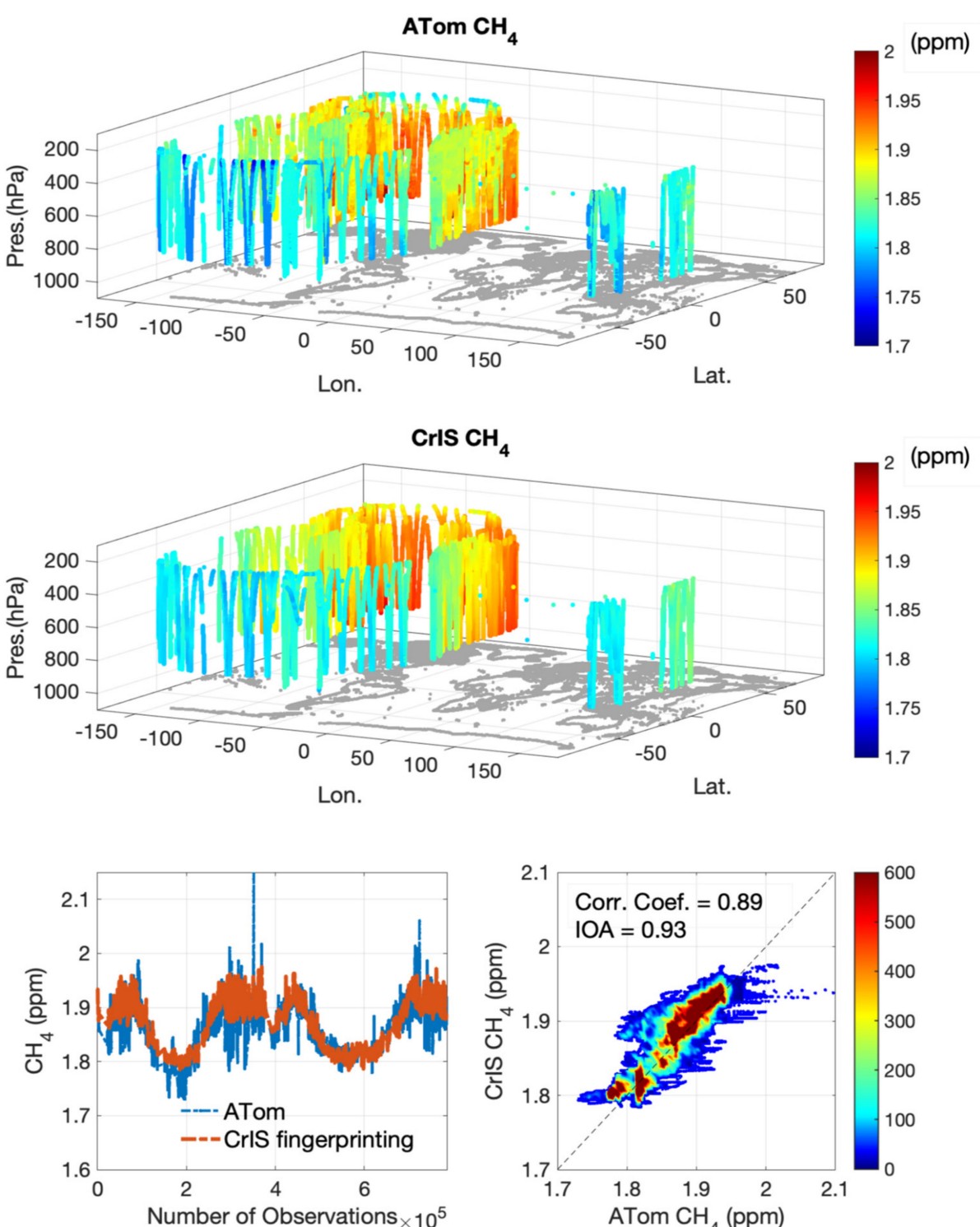

**Figure 7.** Upper panel: 3D illustration of CH$_4$ VMRs obtained via in situ Atom observations from 2017; middle panel: CH$_4$ VMRs retrieved from collocated SNPP CrIS observations; lower left panel: inter-comparison between two sets of results via an overplay plot; lower right panel: scatter plot of CrIS versus ATom CH$_4$ VMRs for all samples, with the number density of the observations being illustrated by the color scale. Similar to Figure 5, the agreement between the fingerprinting results and observations is illustrated using the linear correlation coefficient and IOA. It should be noted here that the total number of collocated samples for 2017 is much larger than that for 2016, leading to the difference in color bar scales between Figures 5 and 6.

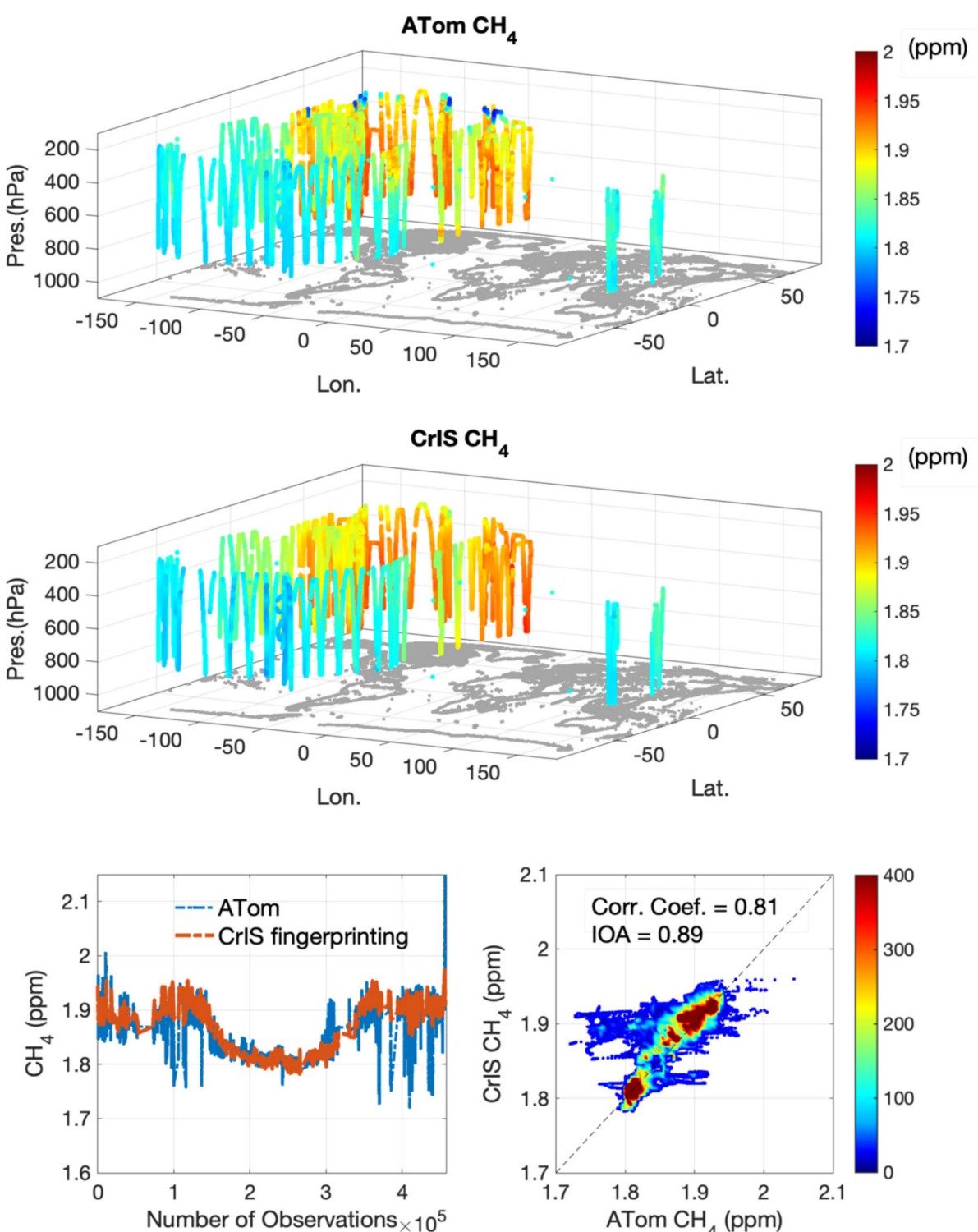

**Figure 8.** Upper panel: 3D illustration of CH$_4$ VMRs obtained via in situ ATom observations from 2018; middle panel: CH$_4$ VMRs retrieved from collocated SNPP CrIS observations; lower left panel: inter-comparison between two sets of results via an overplay plot; lower right panel: scatter plot of CrIS versus ATom CH$_4$ VMRs for all samples measured at a wide range of height and geolocations. The number density distribution of the observations, the correlation, and the index of agreement between CrIS and ATom are also presented.

Statistics regarding the difference between the CrIS fingerprinting and in situ measurement results are detailed in Table 2. Both the bias and the standard deviation values below

300 hPa are around 1% or smaller. The bias of the CrIS—ATom difference is comparable to what was reported in another CrIS—ATom inter-comparison study which used $CH_4$ results from the NOAA-Unique Combined Atmospheric Processing System (NUCAPS) [38]. However, it is noted that our temporal collocation criterion of $\pm 12$ h is much more relaxed compared with the $\pm 1.5$ h criterion used for the NUCAPS $CH_4$ study. Such a choice is based on the consideration that that the change in atmospheric $CH_4$ concentration in the troposphere is predominantly less than 1% (root mean square difference) within a 12 h timeframe, as evidenced by statistics derived from CT-$CH_4$ and CAM reanalysis data. Importantly, our study benefits from a significantly larger sample size that is two orders of magnitude higher than that used in the NUCAPS $CH_4$ study.

**Table 2.** Statistical difference between the CrIS fingerprinting and the ATom $CH_4$. The mean $\pm$ standard deviation values of CrIS—Tom (in relative error %) have been computed for different layers (separated by upper and lower pressure levels). They are also split into regions: Arctic above 60°N, northern mid-latitude between 30°N and 60°N (N-Mid), tropics between 30°S and 30°N, southern mid-latitude between 60°S and 30°S (S-Mid), and Antarctic below 60°S.

| Pressure (hpa) | Arctic | N-Mid | Tropics | S-Mid | Antarctic | Global |
|---|---|---|---|---|---|---|
| above 250 | $5.34 \pm 3.50$ | $1.90 \pm 2.83$ | $0.21 \pm 1.18$ | $0.89 \pm 2.16$ | $0.58 \pm 1.42$ | $1.77 \pm 3.02$ |
| 250–350 | $2.39 \pm 2.66$ | $1.01 \pm 1.50$ | $0.32 \pm 1.02$ | $0.24 \pm 0.92$ | $0.49 \pm 1.35$ | $0.87 \pm 1.71$ |
| 350–450 | $1.13 \pm 1.01$ | $0.93 \pm 0.95$ | $0.46 \pm 1.09$ | $0.37 \pm 0.81$ | $0.07 \pm 0.93$ | $0.71 \pm 1.04$ |
| 450–550 | $0.69 \pm 0.77$ | $0.99 \pm 1.00$ | $0.58 \pm 1.08$ | $0.26 \pm 0.77$ | $0.22 \pm 0.95$ | $0.65 \pm 0.98$ |
| 550–650 | $0.40 \pm 0.91$ | $1.05 \pm 1.07$ | $0.68 \pm 1.09$ | $0.22 \pm 0.71$ | $0.41 \pm 0.85$ | $0.65 \pm 1.03$ |
| 650–750 | $0.45 \pm 0.80$ | $0.91 \pm 1.27$ | $0.67 \pm 1.20$ | $0.33 \pm 0.73$ | $0.42 \pm 0.89$ | $0.63 \pm 1.10$ |
| 750–850 | $0.51 \pm 0.82$ | $0.80 \pm 1.19$ | $0.57 \pm 1.30$ | $0.42 \pm 0.78$ | $0.36 \pm 0.88$ | $0.59 \pm 1.10$ |
| 850–950 | $0.29 \pm 0.80$ | $0.52 \pm 1.52$ | $0.40 \pm 1.34$ | $0.42 \pm 0.90$ | $0.52 \pm 0.96$ | $0.42 \pm 1.22$ |
| below 950 | $-0.04 \pm 1.40$ | $1.33 \pm 1.82$ | $0.27 \pm 1.42$ | $0.40 \pm 1.06$ | $0.30 \pm 0.85$ | $0.61 \pm 1.58$ |

A similar satellite–airborne inter-comparison study validated $CH_4$ results retrieved from the Aura Tropospheric Emission Spectrometer (TES) and AIRS radiances using ATom data [39]. This study implemented a coincident criteria of 9 h and 50 km, closer to the criteria used in our study. Additionally, specific quality control was applied to screen out low-sensitivity and cloudy cases, leading to a low yield rate (~1/4) of samples [39]. The difference between AIRS and in situ results exhibited a bias of ~3% over the ocean and ~4% over land before bias correction, with a standard deviation less than 2% [39]. Both the NUCAPS and the TES-AIRS $CH_4$ validation studies involved the use of averaging kernel correction to take into account the relative difference introduced by averaging kernel 'smoothing' [40]. However, applying averaging kernel correction on ATom data necessitates extending ATom $CH_4$ measurements within the limited vertical region to match the complete profile measured by the sounders. It can be difficult to assess errors introduced by extending aircraft measurements using the assumed 'true' profiles typically obtained from global atmospheric chemistry models [7,10,40]. Nalli et al. [38] cautioned that applying averaging kernel correction for an inter-comparison study between satellite-based and airborne measurements can be misleading under certain conditions when the retrieval system has little to no measurement sensitivity to $CH_4$. Based on these considerations, we did not implement averaging kernel correction in our study. Therefore, the difference between the spectral fingerprinting results and ATom results include the null-space errors.

The error statistics listed in Table 2 reflect the accuracy of the $CH_4$ derived under all-sky conditions without the imposition of a carefully designed quality control scheme to filter out low-sensitivity samples with potentially large errors. Despite the absence of a quality control scheme and the retention of null-space errors, the precision and accuracy of our results are still comparable to, or even better than, the NUCAPS and TES-AIRS results.

This suggests the potential of using spectral fingerprinting methodology to enhance the $CH_4$ data products from existing hyper-spectral sounder missions.

Table 2 shows that the errors in the lower troposphere are most pronounced in the northern mid-latitude region as opposed to the other regions. Compared with the ATom observations, the fingerprinting results for individual CrIS footprints demonstrate a consistently positive bias globally. The most substantial errors, both in terms of bias and standard deviation, can be observed in the upper troposphere to lower stratosphere region (above 250 hPa) over the Canadian wetlands and Greenland regions. These discrepancies can be attributed to the limitations in the vertical resolution of CrIS spectral measurements, preventing the precise sensing of the rapid decrease in methane concentrations in the upper-troposphere to low-stratosphere altitudes in the Arctic region. This challenge is particularly notable in regions where the tropopause height can be as low as 300 hPa.

## 5. Conclusions and Future Work

Compared to ground-based measurements, which serve as anchor observations for long-term $CH_4$ trends and inter-annual variations in the atmosphere, satellite-based measurements play a crucial role in assessing the geographical distribution of $CH_4$ concentrations globally. Accurate information about the sources and sinks of $CH_4$ is essential for climate models to predict atmospheric $CH_4$ concentrations. However, existing $CH_4$ data products based on infrared (IR) sounder observations face limitations in accurately capturing the global geographical distribution characteristics of $CH_4$. A major challenge arises in areas where satellite-based measurements lack information content due to factors like insufficient thermal contrast or cloud blockage. To address this challenge, scene-dependent a priori information is required to enhance $CH_4$ retrieval in these areas. While high-quality $CH_4$ information can be obtained from $CH_4$ data assimilation systems like CT and CAM reanalysis, spectral radiance information from IR sounders has yet to be adequately assimilated in those systems. Additionally, considerations for data latency, especially in applications for environmental monitoring, hinder the direct use of $CH_4$ data from these systems to provide a priori constraints for physical retrievals.

We have developed a spectral fingerprinting scheme to tackle the challenges of retrieving $CH_4$ from satellite-based IR hyper-spectral sounder measurements. The scheme has a lot of similarities to a data assimilation system, but it differs from this type of system because it uses a machine learning-based model to initialize the a priori background and an optimized scheme to enhance the spectral fingerprints of $CH_4$. The fingerprinting algorithm follows a 'lazy learning' methodology to efficiently identify a group of matched $CH_4$ profiles using the optimized CrIS spectral radiances as the predictors. The a priori information that largely retains the accuracy and characteristics of CT can be provided in a real-time (near-real-time) manner via a classification scheme based solely on the spectral radiance measured by sounders. A final solution for the $CH_4$ profile can be obtained by using a radiative kernel-based optimal inversion procedure to fit the optimized spectral radiance signals from CrIS measurements. This combination of machine learning and radiative kernel-based inversion has the potential to offer advantages in terms of accuracy and computational efficiency in the context of sounder-based $CH_4$ retrieval.

We have demonstrated that the $CH_4$ retrieved from SNPP CrIS observations via the fingerprinting method can generally catch the vertical and spatial distribution characteristics at a global scale and at different seasons, using the CAMS $CH_4$ reanalysis data as the reference. A validation study carried out using in situ ATom data demonstrated that both the systematic error and the uncertainty associated with the derived $CH_4$ profiles at various altitudes in the tropospheric region range from less than 1% to no more than 2%.

The fingerprinting scheme leverages SiFSAP to generate radiative kernels under all-sky conditions. The results and error statistics presented herein are associated with individual CrIS observations under both clear- and cloudy-sky conditions. Quality temperature, water vapor, surface, and cloud properties from the SiFSAP used in the offline training ensure the

accuracy of the radiative kernels at the individual footprint scale of the CrIS observations, thereby safeguarding the accuracy of the $CH_4$ fingerprinting at the fine spatial–temporal scale.

This paper has demonstrated the potential of using satellite-based spectral observations to facilitate the instantaneous monitoring of height-resolved methane distribution. This study showcases the effectiveness of employing machine learning to overcome the challenges posed by modeling and measurement errors in a standard retrieval scheme. Our future work will focus on enhancing the accuracy of the a priori background state by exploring more sophisticated machine learning models. Additionally, efforts will be made to integrate fingerprinting results as the a priori information for the iterative physical retrieval procedure used for the production of SiFSAP $CH_4$.

**Author Contributions:** Conceptualization, W.W. and X.L.; methodology, W.W., X.L. and Q.Y.; software, W.W.; validation, W.W.; formal analysis, W.W.; investigation, W.W. and X.L.; resources, X.L. and L.Z.; writing—original draft preparation, W.W.; writing—review and editing, W.W., X.L., X.X., Q.Y., L.L., L.Z., D.K.Z. and A.M.L.; project administration, X.L.; funding acquisition, X.L. All authors have read and agreed to the published version of the manuscript.

**Funding:** This research was funded by the NASA 2017 Research Opportunities in Space and Earth Sciences (ROSES) solicitation (grant number NNH17ZDA001N-TASNPP) and the NASA 2020 ROSES solicitation (grant number NNH20ZDA001N-SNPPSP).

**Data Availability Statement:** Data from the ATom campaign are located at https://doi.org/10.3334/ORNLDAAC/1925 (accessed on 2 September 2023) [41]. The Carbon Tracker CT-CH4-2023 results were provided by NOAA Global Monitory Laboratory (GML), Boulder, CO, USA, from the following website: https://gml.noaa.gov/ccgg/carbontracker-ch4/ (accessed on 11 October 2023) [42]. The SiFSAP data used to generate radiative kernels can be obtained from the NASA Goddard Earth Sciences Data and Information Services Center (GES DISC) at https://disc.gsfc.nasa.gov/datasets/SNDRSNIML2SFSPSUP_2/summary?keywords=SiFSAP (accessed on 1 December 2023).

**Acknowledgments:** We really appreciate Youmi Oh of NOAA GML for providing the CT-CH4-2023 data.

**Conflicts of Interest:** Authors Qiguang Yang and Liqiao Lei are employed by the ADNET Systems Inc. The remaining authors declare that the research was conducted in the absence of any commercial or financial relationships that could be construed as a potential conflict of interest.

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
