# Peer review of "Spectral Fingerprinting of Methane from Hyper-Spectral Sounder Measurements Using Machine Learning and Radiative Kernel-Based Inversion"

_remotesensing, doi:10.3390/rs16030578_

Round 1
Reviewer 1 Report
Comments and Suggestions for Authors
Minor suggestions
a) Consider a change to the title
Spectral fingerprinting of methane from CrIS measurements using machine learning and radiative kernel-based inversion.
b) The Abstract could be improved by highlighting the advantages and limitations of the proposed method, as well as the implications and applications of the results for global CH4 monitoring.
Reviewer 2 Report
Comments and Suggestions for Authors
Figure 1 includes solid blue dots and red squares that are not explained in the legend of figure caption. Please add these elements to a legend explaining that they are shoulder or valley channels.
Table 1 caption should clarify that the shoulder and valley channels are for the dBT/dCH4 spectra.
Figure 3 caption should be completely spelled out. The reader should be able to read and understand all pertinent information from the figure caption and not need to see another caption.
Figure 4 should clarify whether the latitudinal distribution is the average of all longitudes, or for a range of longitudes.
Figure 5/6/7 the scale bars on the right should include the units. Bottom right plot of CH4 and observations should increase the size of the legend dot size so that it is clear which trendline is ATom vs CrIS. Lower right plot should clarify what the R value refers to in the figure caption. The R value also appears to have a black dot beside it which does not appear in the plot. It should be clear that the color indicates the sample density in terms of the number of observations. I would also suggest using the Index of Agreement instead of a correlation or linear model. The scale of the lower right plot changes between figures. While I agree that showing the density of observation points via color is the most important, the authors should make more of an effort to highlight this differences in either the figure caption or the plots.
Reviewer 3 Report
Comments and Suggestions for Authors
The study deals with the retrieval of atmospheric CH4 profiles using a novel spectral fingerprinting methodology and demonstrate performance using measurements from the CrIS sensor aboard the Suomi National Polar-orbiting Partnership (SNPP) satellite. The spectral fingerprinting methodology uses optimized CrIS radiances to enhance the CH4 signal and a machine learning classifier to constrain the physical inversion scheme.
The reviewer recommends minor revision before the manuscript is accepted for publication in Remote Sensing. Some of the concerns/ comments and suggestions to improve the manuscript are given below.
1. The latest research progress in the literature review section is relatively scarce. It is recommended to consider introducing more recent related research, especially advanced technologies and theories in the fields of spectral analysis and environmental monitoring.
2. The authors should compare and analyze the results from different data sources, as well as the similarities, differences, and advantages and disadvantages with other existing methods.
3. The authors did not provide a detailed explanation of the source, coverage, and quality control process of the data. Furthermore, it is recommended to elaborate on whether the uncertainty and error of data from different sources have been taken into account in the comparative analysis.
4. The author's analysis in the discussion section is not in-depth enough, especially regarding the potential impact on climate change monitoring and environmental policy formulation.
Reviewer 4 Report
Comments and Suggestions for Authors
Authors describe the retrieval of atmospheric CH4 profiles using a proposed spectral fingerprinting methodology and demonstrate its performance on satellite sensors measurements. The spectral fingerprinting methodology uses optimized CrIS radiances to enhance the CH4 signal, and a machine learning classifier was applied as well. this seems to be an interesting work. But I think its should be improved before considering for publication. A few issues are raised as follows,
I am curious about the satellite raw data. would you please give a rough overview on the data and display necessary descriptive statistics.
Please clarify the mechanism of the spectral fingerprinting methodology, especially explain the formulas at their solo terms.
In section 3.3, please give detail explanations for the constraints on CH4. Why can they be explicated by the formulae.
Please explain the mask or the filter you used to generate Figures 5-7.
Please use -1 as the superscript for cm-1.
Comments on the Quality of English Languagemoderate revision is needed.
